# Co-Seismic Ionospheric Disturbances Following the 2016 West Sumatra and 2018 Palu Earthquakes from GPS and GLONASS Measurements

Mokhamad Nur Cahyadi [1,2,*], Buldan Muslim [3], Danar Guruh Pratomo [1], Ira Mutiara Anjasmara [1], Deasy Arisa [4], Ririn Wuri Rahayu [1], Irena Hana Hariyanto [1], Shuanggen Jin [5] and Ihsan Naufal Muafiry [6]

[1] Geomatics Engineering Department, Institut Teknologi Sepuluh Nopember (ITS), Surabaya 60111, Indonesia; guruh@geodesy.its.ac.id (D.G.P.); ira@geodesy.its.ac.id (I.M.A.); ririn.19033@mhs.its.ac.id (R.W.R.); irena.hana16@mhs.geodesy.its.ac.id (I.H.H.)

[2] Research Center for Science-Technology of Marine and Earth, Directorate of Research and Community Service Institut Teknologi Sepuluh Nopember, Surabaya 60111, Indonesia

[3] National Institute of Aeronautics and Space, LAPAN, Bandung 40173, Indonesia; buldan.muslim@lapan.go.id

[4] Research Center for Geotechnology, Indonesian Institute of Sciences, BRIN, Bandung 40135, Indonesia; deasy.arisa@lipi.go.id

[5] Shanghai Astronomical Observatory, Chinese Academy of Sciences, Shanghai 200030, China; sgjin@shao.ac.cn

[6] Survey and Mapping Department, Sinar Mas of Polytechnic Berau Coal, Kecamatan Tanjung Redeb, Berau 77315, Indonesia; ihsan@polteksimasberau.ac.id

*   Correspondence: cahyadi@geodesy.its.ac.id

**Abstract:** The study of ionospheric disturbances associated with the two large strike-slip earthquakes in Indonesia was investigated, which are West Sumatra on 2 March 2016 (Mw = 7.8), and Palu on 28 September 2018 (Mw = 7.5). The anomalies were observed by measuring co-seismic ionospheric disturbances (CIDs) using the Global Navigation Satellite System (GNSS). The results show positive and negative CIDs polarization changes for the 2016 West Sumatra earthquake, depending on the position of the satellite line-of-sight, while the 2018 Palu earthquake shows negative changes only due to differences in co-seismic vertical crustal displacement. The 2016 West Sumatra earthquake caused uplift and subsidence, while the 2018 Palu earthquake was dominated by subsidence. TEC anomalies occurred about 10 to 15 min after the two earthquakes with amplitude of 2.9 TECU and 0.4 TECU, respectively. The TEC anomaly amplitude was also affected by the magnitude of the earthquake moment. The disturbance signal propagated with a velocity of ~1–1.72 km s$^{-1}$ for the 2016 West Sumatra earthquake and ~0.97–1.08 km s$^{-1}$ for the 2018 Palu mainshock earthquake, which are consistent with acoustic waves. The wave also caused an oscillation signal of ~4 mHz, and their azimuthal asymmetry of propagation confirmed the phenomena in the Southern Hemisphere. The CID signal could be identified at a distance of around 400–1500 km from the epicenter in the southwestern direction.

**Keywords:** earthquake; GNSS; acoustic wave; co-seismic ionospheric disturbances (CIDs)

## 1. Introduction

GNSS can estimate the ionospheric total electron content (TEC) by integrating several electron densities along the line-of-sight (LOS) between the receiver and the satellite [1,2]. This measurement has been used in various kinds of disturbances, e.g., mine blasts [3], geomagnetic storms [4], volcanic eruption [5,6], and ballistic missiles [7]. In particular, many studies of the ionospheric electron density variation influenced by earthquakes were conducted, and some co-seismic ionospheric disturbances (CIDs) were observed, which provided some insights on earthquakes [8–12].

A large earthquake of Mw 7.8 occurred on 2 March 2016, which was located in the Indian Ocean with a distance of approximately 600 km from the southwest of the

Sumatra coast, where the Indo-Australian plates subduct north–northeastward beneath the Sundaland plate. This earthquake caused a large slip in the fault of 185 × 20 km. An Mw 8.6 intraplate earthquake and Mw 8.2 aftershock also occurred in the Indian Ocean on 11 April 2012, with the similar left-lateral strike-slip focal mechanisms to the earthquake on 2 March 2016. On 28 September 2018 at 10:02 UT, an earthquake occurred in the city of Palu with a magnitude of Mw 7.5, whose epicenter was located about 50 miles north of the city of Palu, around 0.256° S in latitude and 119.846° E in longitude. This earthquake occurred as a result of a fault at a shallow depth inside the micro plate of the Maluku Sea, part of the wider Sunda tectonic plate.The type of this earthquake that occurs is strike-slip with typically ~120 × 20 km in size. This mainshockcaused the appearance of a tsunami with a height of 3.8 m. The tsunami struck towards the land about 6–12 min after the earthquake happened [13]. Based on the survey from UNESCO's International Tsunami Survey Team on the 125 km long Palu Bay coastline to the epicenter area, it was found that the tsunami height reached up to 9.1 m, and the inundation height reached 8.7 m [14]. Goda et al. [15] stated that strike-slip events were able to trigger a huge tsunami from the 2018 Palu earthquake. The co-seismic defamation, submarine landslides, and tidal effects were also important factors in generating the tsunami in Palu Bay. Recently, the CID in the strike-slip earthquake has been analyzed in detail by Cahyadi and Heki [16], but it was limited to the earthquakes with significant magnitude moments only. The CID Sumatra earthquake on 11 April 2012 was investigated, with the seismic moment magnitude (Mw) of 8.6 and 8.4. Evaluation of the correlation is needed using a strike-slip earthquake with a lower moment magnitude.

The study of CID on the strike-slip earthquake in Sumatra is still very limited, since the usual earthquake happened with the uplift or the normal type. Several CID research studies have been conducted for earthquakes that happened in Sumatra, including the 2004 Sumatra–Andaman earthquake, 9.1 Mw [17–19]; the 2005 Nias earthquake, 8.6 Mw [20,21]; the 2007 Bengkulu earthquake, 8.5 Mw [21]; and the 2012 North Sumatra Earthquake with an 8.6 Mw mainshock and 8.2 Mw aftershock [16]. The recorded strike-slip earthquake that happened in Sumatra is the 2000 South Sumatra earthquake with 7.9 Mw, and the 2012 North Sumatra earthquake with 8.6 Mw and 8.2 Mw. Cahyadi et al. [22] conducted research about CIDs of the 2016 West Sumatra earthquake, but it was mainly focused on the analysis of time occurrence and the magnitude of the CID anomaly on the location of the receiving station.

Further research about the 2018 Palu earthquake was explained in detail by Liu et. al. [23], which was focused on the utilization of tsunami traveling ionospheric disturbances (TTIDs) to obtain the source of earthquakes using the data from the International GNSS Service (IGS). As a result, the information regarding the earthquake's epicenter was obtained from the velocity and the distance between signal propagation from the epicenter. On the other hand, Mikesel et al. [24] analyzed the CID of the 2018 Palu earthquake using IonoSeis software with the IGS station and used the International Reference Ionosphere (IRI2016) model to compute the background electron density (Ne0). Furthermore, this paper explains the CID analysis of the 2018 Palu earthquake using the data from Indonesian GPS stations operated by BIG (Indonesian Geospatial Agency), and the results are compared to the previous studies. Additionally, comprehensive analysis of the tsunami during the 2018 Palu earthquake is carried out by adding bathymetry survey data from the Palu Bay. The data will be used to determine changes in material volume caused by tsunamis.

In this paper, the CIDs of both earthquakes is investigated using INACORS (Indonesia Continuously Operating Reference Station) and SUGAR stations. Considering the strike-slip mechanism of the 2016 West Sumatra earthquakes and the 2018 Palu earthquake, it would be interesting to compare the CID with an empirical relationship of CID amplitudes of earthquake magnitudes [16]. The 2016 West Sumatra earthquake was investigated by Cahyadi et al. [22], but that study only focused on CID amplitude. Here, the process results of CIDs for the 2016 West Sumatra and the 2018 Palu earthquake are presented with the

focal mechanisms, and the effect between the moment magnitude and the focal mechanism of earthquakes on CID waveforms is discussed.

## 2. Data and Methods

GPS satellites are located ~20,000 km above the Earth's surface. These satellites propagate electromagnetic signal through the ionosphere before being received by GPS receivers on the Earth. GPS satellites transmit three carrier waves, namely L1, L2, and L5 bands (L1 = 1.2, L2 = 1.5 GHz, and L5 = 1.151 GHz), and GLONASS satellites have same the formation, which propagates the band frequencies G1 (1.6 GHz) and G2 (1.3 GHz). For accurate positioning, the ionospheric delay is removed by an ionosphere-free linear combination of the two carrier phases. For ionospheric studies, we derive TEC from the differences of the phases at the two frequencies (L1,L2 for GPS and G1,G2 for GLONASS) as

$$\Delta STEC = \frac{1}{40.308} \times \frac{f_1^2 f_2^2}{(f_1^2 - f_2^2)} \times (\rho_1 - \rho_2) \tag{1}$$

where $f_1$ and $f_2$ are the carrier phase frequency, $\rho_1$ and $\rho_2$ represent the carrier phases from $f_1$ and $f_2$, respectively, and the phase difference $(\rho_1 - \rho_2)$ is expressed with the unit of meters [25]. TEC shows the variation in the ionosphere due to movement of satellites in the sky. This variation also changes, influenced by diurnal variations of the sun's peak angle and long-term disturbances such as large-scale ionosphere travel disturbances (LSTID). High-pass filters are often used for eliminating long-term variation. CIDs were obtained from the differences between observation data and model data. Observation data were obtained using Formula (1) from L1, L2 measurement, while model data were obtained from 6-degree polynomial calculation. A CID is an anomaly of the residual between observation and model data. As explained in Heki [25], the best degree depends on the width of the time window, and the degree is tuned so that the polynomial precisely removes long-period fluctuations.

The data were obtained from the permanent tracking dual-frequency GNSS stations in the InaCORS and SUGAR Array around the 2016 West Sumatra earthquakes (Figure 1a) and the 2018 Palu earthquake (Figure 1b), whose sampling interval is 30s. To investigate the spatial characteristics of a CID, e.g., the propagation velocity of the disturbance, calculation of the point (IPP) of the line of sight was done by assuming the ionosphere as a thin layer with a height of ~300 km in this study. The IPP is projected to the ground and forms a sub-ionosphere (SIP) point. This SIP is often located at around 1500 km from the epicenter, depending on the elevation angle.

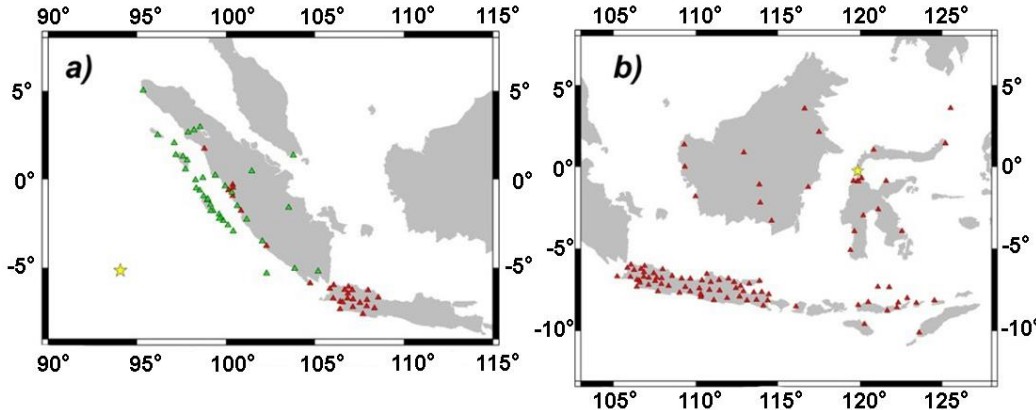

**Figure 1.** GNSS stations from SUGAR array (green triangle) and INACORS (red triangle) used in (**a**) the 2016 West Sumatra earthquake, and (**b**) the 2018 Palu earthquake. The epicenters are marked with the yellow stars. Both were strike-slip earthquakes.

## 3. Results and Analysis

### 3.1. Co-Seismic Ionospheric Disturbances

First, we estimated the CID of the 2016 West Sumatra earthquake (Mw 7.8) with the strike-slip mechanism. About 11–16 min after the earthquake mainshock, the CID was recorded by GPS PRN 9,17,23 and GLONASS PRN 4 satellite with maximum amplitudes of 2.9 TECU and a period of around 4–5 min (Figure 2b). To eliminate long-term variations and isolate the disturbance signal of CIDs, the polynomials of time was applied with the degree up to 6. Residuals from these polynomials are used to study disturbance signal characteristics. Both GPS and GLONASS satellites observed the disturbances similar in waveform, but the amplitudes from GPS PRN 9,17, and 23 were larger.

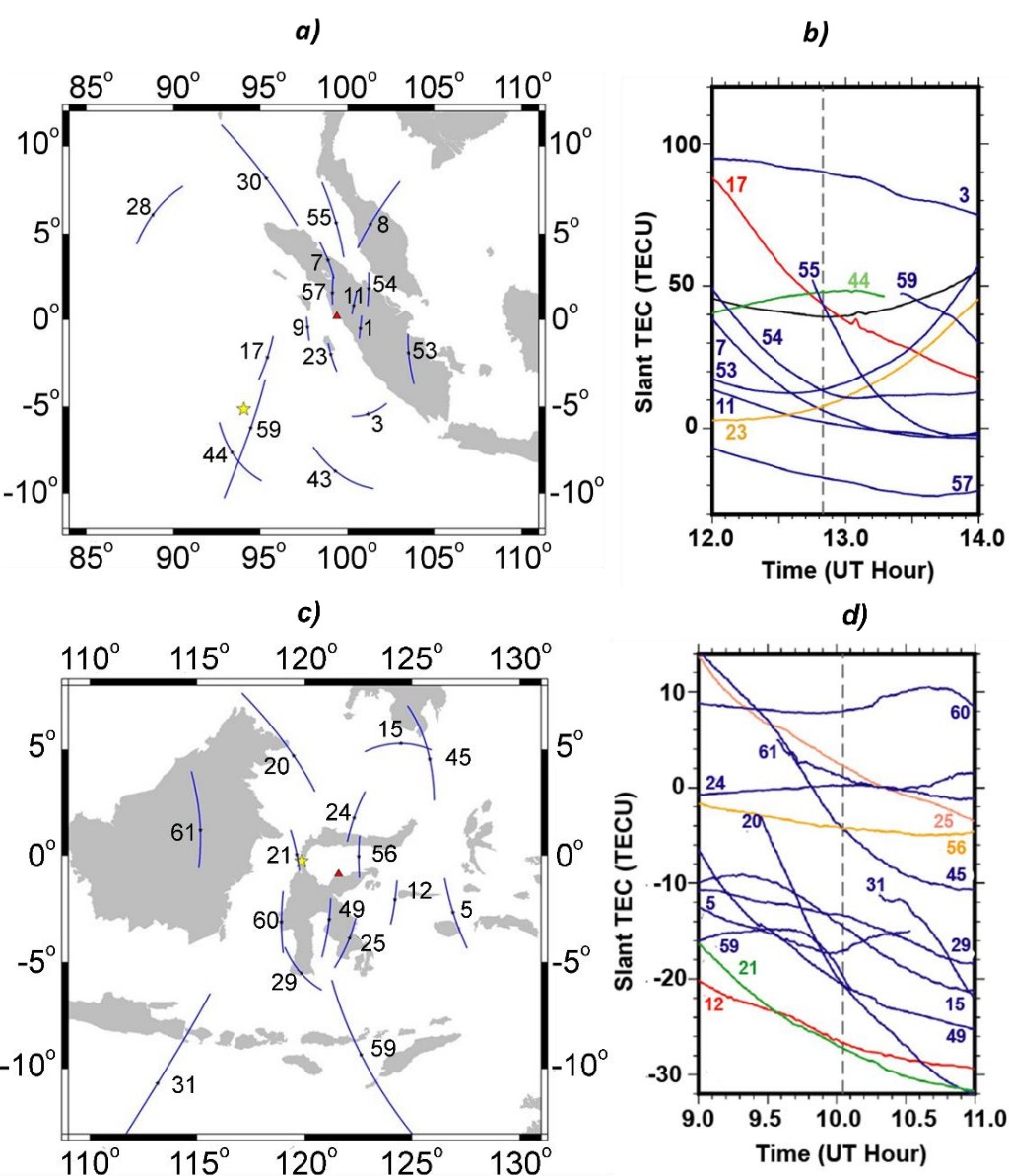

**Figure 2.** *Cont.*

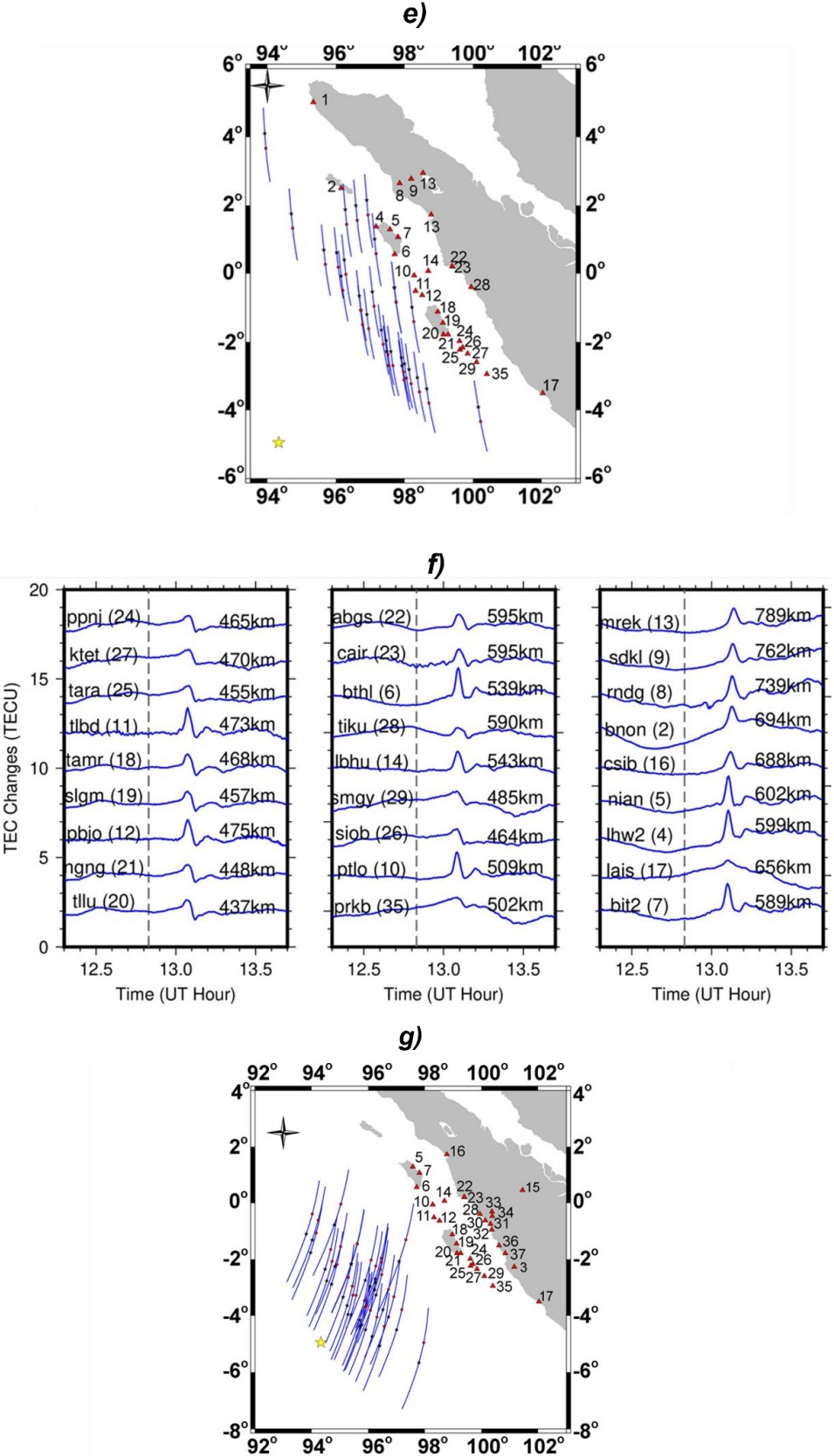

**Figure 2.** *Cont.*

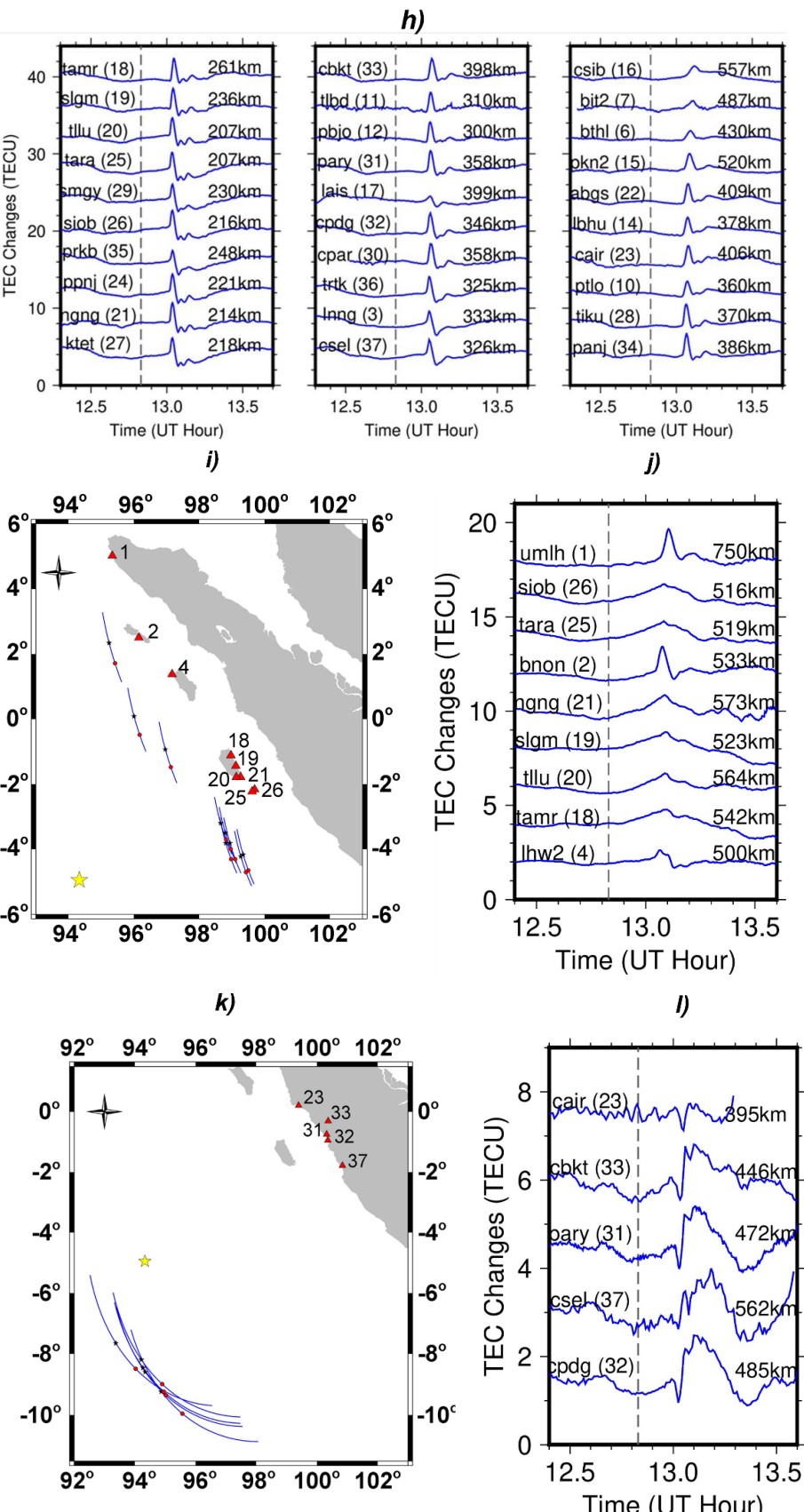

**Figure 2.** *Cont*.

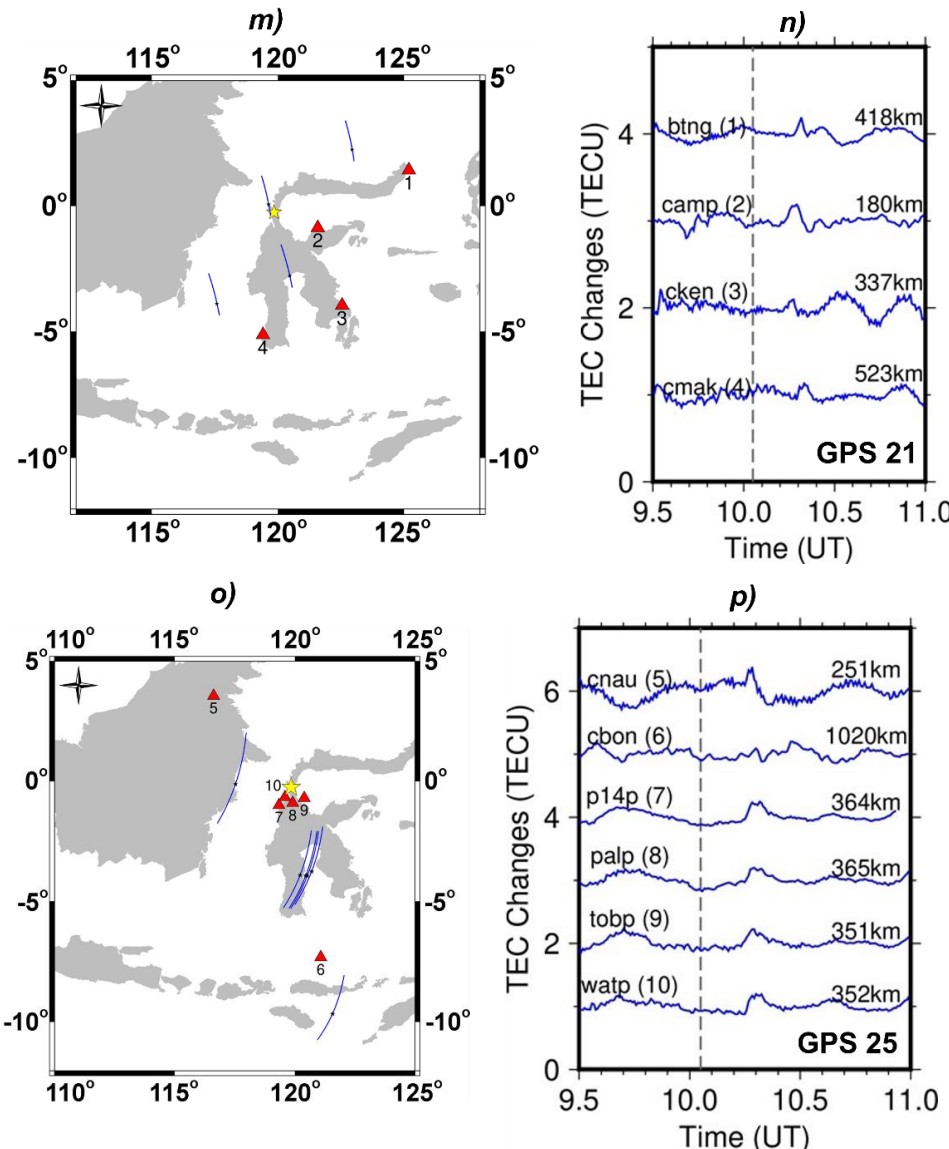

**Figure 2.** Trajectories of SIPs during the 2016 West Sumatra earthquake (**a**) and 2018 Palu earthquake (**c**). Time series changes in TEC for the 2016 West Sumatra earthquake at 12.4–13.6 UT by all satellites (**b**), GPS PRN 9 (**e,f**), 17 (**g,h**), 23 (**i,j**) and GLONASS PRN 4 (**k,l**), and also for the 2018 Palu earthquake at 9.5–11 UT by all satellites (**d**), and GPS PRN 21 (**m,n**) and 25 (**o,p**). The dashed vertical black line represents the time of the earthquake (the 2016 Sumatra earthquake is 12:49:48 UT and the 2018 is 10:02:45 UT), a large yellow star indicates the location of the epicenter of the earthquake, and the small black star on the trajectory indicates the location of the SIPs during the mainshock event. The comparison of the 2016 Western Sumatra earthquake and the 2018 Palu earthquake is shown in Table 1. The numbers inside the bracket at the time series (**f,h,j,l,n,p**) indicate station position.

**Table 1.** Earthquakes data.

| Earthquake Data | 2016 West Sumatra Earthquake | 2018 Palu (Mainshock) Earthquake |
|---|---|---|
| Magnitude | 7.8 Mw | 7.5 Mw |
| Time | 12.83 UT (hour) | 10.046 UT (hour) |
| Epicenter | 94.33° N, −4.952° E | 119.846° N, −0.256° E |
| Depth | 24 km | 20 km |
| Uplift | 1.86 m | 1.04 m |
| Max. CID | 2.9 TECU | 0.4 TECU |

**Table 1.** *Cont.*

| Earthquake Data | 2016 West Sumatra Earthquake | 2018 Palu (Mainshock) Earthquake |
|---|---|---|
| Satellite | GPS PRN 17 | GPS PRN 25 |
| GIM—VTEC | 54.894 TECU | 19.775 TECU |
| Normalized CID | 5.3% | 2% |

The larger CID recorded by GPS PRN 9 (Figure 2f), 17 (Figure 2h), and 23 (Figure 2j) would reflect shallower angles between the line-of-sight and the wave front. GLONASS PRN 4 is in the southern sky, and CID amplitudes are considerably small in the stations to the south of the epicenter (Figure 2k). In the geometry of GLONASS PRN 4, the line of sight (LOS) penetrates the wave front in a deep angle, and the positive and negative electron density anomalies tend to cancel each other [21]. Moreover, in the northern side, the epicenter results in a larger CID. Furthermore, since the SIPs of PRN 9 (Figure 2e), 17 (Figure 2g), and 23 (Figure 2i) are in the northern area of the epicenter, it showed a bigger CID compared to the other CIDs observed by GLONASS PRN 4 [16].

In the case of the 2018 Palu earthquake, the data were obtained from the INACORS station in Sulawesi, Kalimantan, and Java Islands. CIDs appeared after the mainshock and were detected by GPS PRN 21 and 25 with maximum amplitudes of 0.4 TECU, as shown in Figure 2d.

Mikesel et al. [24] calculated the CID of the 2018 Palu earthquake at the BTNG station by comparing TEC observation and its modelling by IRI to identify the ionosphere model. This research conducted a similar calculation of ionospheric linear combination to identify CIDs and their modelling using a polynomial degree up to 6. The results showed that SIP and TEC anomalies were located on the trajectory of satellite 21, which was calculated based on 419 km from the epicenter, and the TEC anomaly was 0.1 TECU. The calculation was added from various stations using INACORS data. This result is similar to the previous one from Mikesell et al. [24], which can be seen in the Appendix A (Figures A1 and A2).

The research focused on four satellites (GPS PRN 9,17, and 23 and GLONASS PRN 4) for the 2016 West Sumatra and 2018 Palu earthquakes (GPS PRN 21 and 25) because the CID is consistent in appearance. The condition of one-direction geometry between the epicenter and GPS receivers can be found in the GPS PRN 9 (Figure 2e), 17 (Figure 2g), and 23 (Figure 2i). This condition is important because it enables shallow LOS penetration with the CID wave front. In order to obtain a clear anomaly, the receiver should be located on the same side of the epicenter as the SIP and farther from the epicenter than the SIP [21].

Figure 2k,l is the CID in the 2016 West Sumatra earthquake observed by the GLONASS PRN 4 satellite. The figure shows a negative anomaly, which was caused by the north directivity of the propagation signal [17]. The epicenter was located in the south of the geomagnetic field alongside its SIP. However, the propagation was headed toward north direction, resulting in a smaller CID [16]. This satellite contained a positive anomaly after the negative CID. We indicated the anomaly as an Equatorial Ionization Anomaly (EIA), where this satellite's SIP at that time was in orbit of the EIA area, confirmed using Global Ionospheric Maps (GIMs). This phenomenon usually occurs at low and equatorial latitudes in the ionosphere, as shown in Figure A3 [26,27].

Based on the Appendix A (Figures A4 and A5), the slant changes of TEC observed by GPS PRN 25 during the 2018 Palu earthquake and GPS PRN 17 during the 2016 Sumatra earthquake showed a moderate variation one day prior to and one day after the earthquake. The exception was during the earthquake day, in which the TEC deviated by more than 3 σ from the model. In order to facilitate further investigation, we provided a table (Table 1) and provided key quantities. We obtained the background vertical TEC at the time and place where the CID were detected using GIM [28].

The CID of the 2018 Palu earthquake could be detected by GLONASS PRN 21 and 25, and also by oscillation resonance (Figure 3a), GPS PRN 21 (BTNG station).

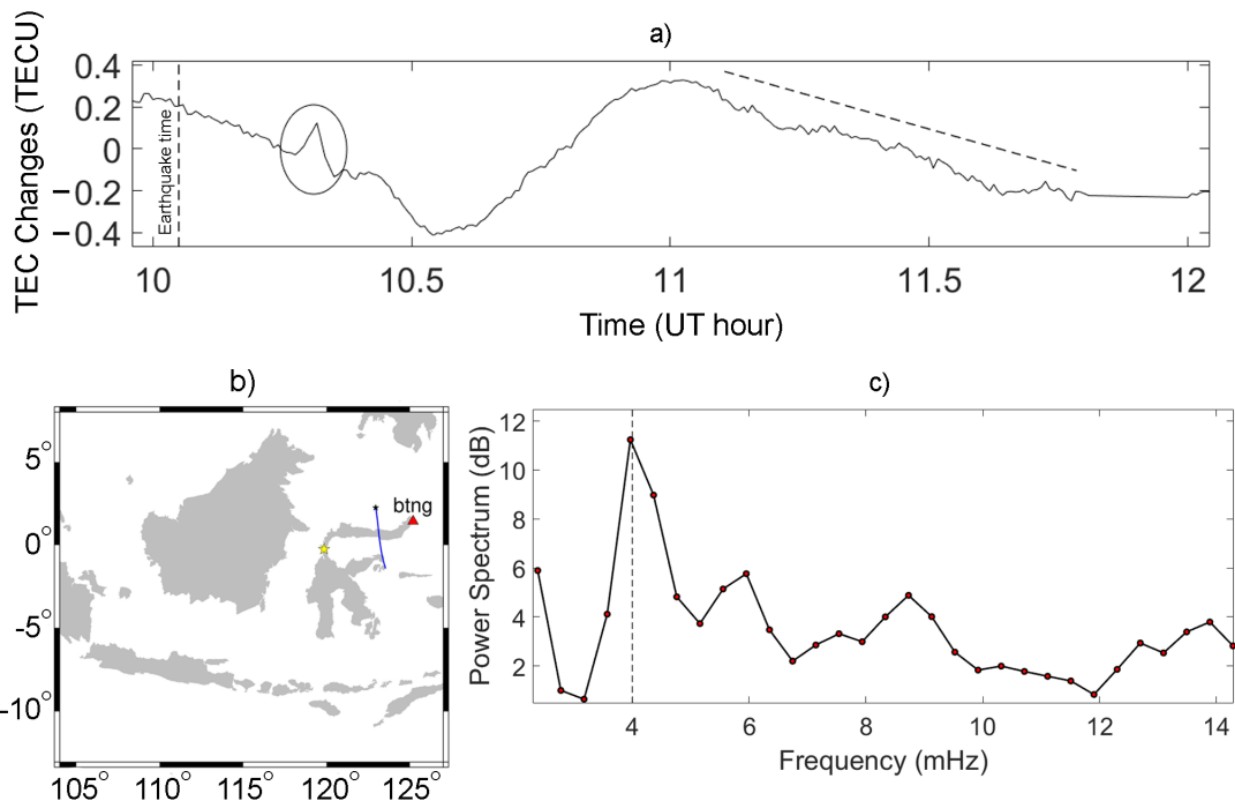

**Figure 3.** (**a**) The high-pass filtered STEC time-series on BTNG (GPS PRN 21) (**b**) SIPs of the satellites, which was observed from the BTNG station. The spectrogram (**c**) shows a peak around 4 mHz, which is the dominant atmospheric modes that trigger resonant coupling with the solid Earth shown by the dashed line [29]. The vertical, slant, and circle lines in (**a**) represent the time of the earthquake (10:02:45 UT), time windows for the spectral analyses, and CID, respectively. The vertical dashed lines in (**c**) are the frequency peaks at 4 mHz.

Figure 3 shows the spectrogram obtained by the Blackman–Tukey method using the TEC time-series after the mainshock. The observed peak frequencies were ∼4 mHz. This is consistent with the atmospheric resonance frequencies during certain time windows (~1 h). In order to clarify the dominant frequency, the BTNG station was chosen to calculate power spectral density with the significant oscillations ranging about 55 min (11.1–11.8 UT). This time range produced a frequency at 4 mHz as the dominant frequency. This frequency is one of the atmospheric resonance frequencies that often appear after large earthquakes [6,18,29–34]. In this result, one way would be to give the range of power about ±0.5 of the peak power and frequency about ±0.3 mHz. The sampling data was 30 s, while the maximum frequency in the periodogram was 16.67 mHz. This value is in accordance with the results obtained from 2004 Sumatra–Andaman earthquake [18], the 2011 Tohoku-Oki earthquake [30,31], and the 2007 Bengkulu earthquake [21]. Resonance in the atmosphere is caused by acoustic waves generated by movements in the lithosphere (earthquake) that propagate vertically. This wave is reflected downward and interferes with the wave propagating upward; this phenomenon causes resonant oscillations [35,36]. Oscillations were clearly detected after the 2018 Palu earthquake from station BTNG, which was observed by satellite 21 with a distance about 700 km north of the epicenter, respectively.

### 3.2. Propagation Speed

CIDs are caused by three different atmospheric waves, direct acoustic waves from the epicenter 10–15 min after the earthquake as fast as ∼0.6–1.0 km s$^{-1}$. Heki and Ping [17] found that the CID of the 25 September 2003 (19:50:06 UT) Tokachi-Oki Earthquake (Mw 8.0) had two separate components with different propagation speeds, which were 3.8 km s$^{-1}$

and 1.2 km s$^{-1}$. The faster velocity (~4 km s$^{-1}$) was excited by the Rayleigh surface waves and propagated farther than those by direct acoustic waves [37]. The last atmospheric waves were gravity waves, caused by large tsunamis with velocity (~0.3 km s$^{-1}$) [12].

Figure 4a–c show that the propagation velocity of CIDs observed by GPS PRN 9, 23, and 17 was about 1.29 km s$^{-1}$, 1.1 km s$^{-1}$, and 1.42 km s$^{-1}$, respectively, and it was observed by the maximum peak to peak. On the other hand, satellite GLONASS PRN 4 (Figure 4d) also observed the velocity of signal propagation, which was about 1.72 km s$^{-1}$. These propagations are consistent with acoustic wave velocities. On the other hand, a CID of the 2018 Palu earthquake was observed by GPS PRN 21 and 25. The travel-time diagram for the satellite is shown in Figure 4e,f where the CID occurred ~13 min after the mainshock with a propagation speed of about 0.97–1.08 km s$^{-1}$. This propagation is consistent with the acoustic wave. The apparent velocities showed Pearson's correlation about 0.73–0.99, and the one sigma errors were about 0.01–0.53, respectively. These show that the accuracies of velocities were quite good in sigma errors and Pearson's correlation. The highest accuracy error happened at GLONASS PRN 4 ($\pm$0.53 km/s) because of the EIA phenomenon that occurred, causing fluctuations. However, the error did not change the type of wave propagation. The acoustic wave appeared in this apparent velocity. We could not find a gravity wave of 0.3 km s$^{-1}$ clearly that would indicate the presence of a tsunami wave. Based on Liu et al. [23], the effect of the 2018 Palu earthquake recorded by an Australian IGS station, namely kat1, reached ~2500 km in height with an average velocity of 228 m s$^{-1}$. The tsunami occurred 25 min after the mainshock. The wave created a big impact toward ionosphere anomaly (red line) with propagation velocity ~0.3 km s$^{-1}$ [38]. Another measurement was done to find out the epicenter location from distance and speed modelling of ionospheric propagation anomalies based on the data in Indonesia. The result is similar with Liu et al. [23], which can be seen in the Appendix A (Figures A6 and A7).

*3.3. Relation with the Focal Mechanism*

The focal mechanism of an earthquake describes the estimation of the three-dimensional configuration of the lithosphere [39]. The components have been derived from a solution of the moment tensor for the earthquake, which itself is estimated by an analysis of observed seismic waveforms [36]. Near field CIDs are excited by co-seismic vertical crustal movements. Figure 5 shows such movements calculated using fault parameters inferred seismologically. A numerical solution was used for strike-slip-dip faults and Green's function for an elastic half-space [40], assuming a uniform rigidity of 50 Gpa. The CIDs that were found in the 2016 West Sumatra earthquake would have been excited by vertical movement of the ocean floor (and hence sea surface), as shown in Figure 5a. The strike-slip earthquake has a vertical component, and this caused a positive change in the TEC anomaly. Figure 5a,b shows an uplift component of 1.8 m and 1.04 m respectively, even though the earthquake mechanism was strike-slip.

The 7.5 Mw Palu earthquake occurred on 28 September 2018 at 10:02 UT, with the earthquake epicenter centered on the coast of Sulawesi Island, about 50 miles north of the city of Palu. Based on USGS data, parameters of dip, strike, and slip were 67°, 350°, and −17°, respectively. The results of Okada modeling for the earthquake are shown by Figure 5b. The earthquake mechanism was a strike-slip earthquake, while the CID response to this earthquake showed an inverted N wave shape, seen in Figure 2c (inverted N-type wave) [37]. The vertical movement of the ocean floor (and sea surface as well) triggered the CIDs discovered in this study, as presented in Figure 5a for the 2016 West Sumatra earthquake and Figure 5b for the 2018 Palu earthquake. Even a pure strike-slip earthquake can produce vertical crustal movements to some extent [16]. The TEC responses to the 2018 Palu earthquake were where the co-seismic vertical movements were dominated by subsidence. The co-seismic TEC changes observed by GPS PRN 25 showed a compound signal, an inverted N-wave (Figure 2o), that was detected until ~1400 km. On the other hand, the apparent velocity of the negative disturbance was ~0.97 km s$^{-1}$ (Figure 4f). Thus,

our observations suggest that ground subsidence induces ionosphere disturbances starting with negative changes.

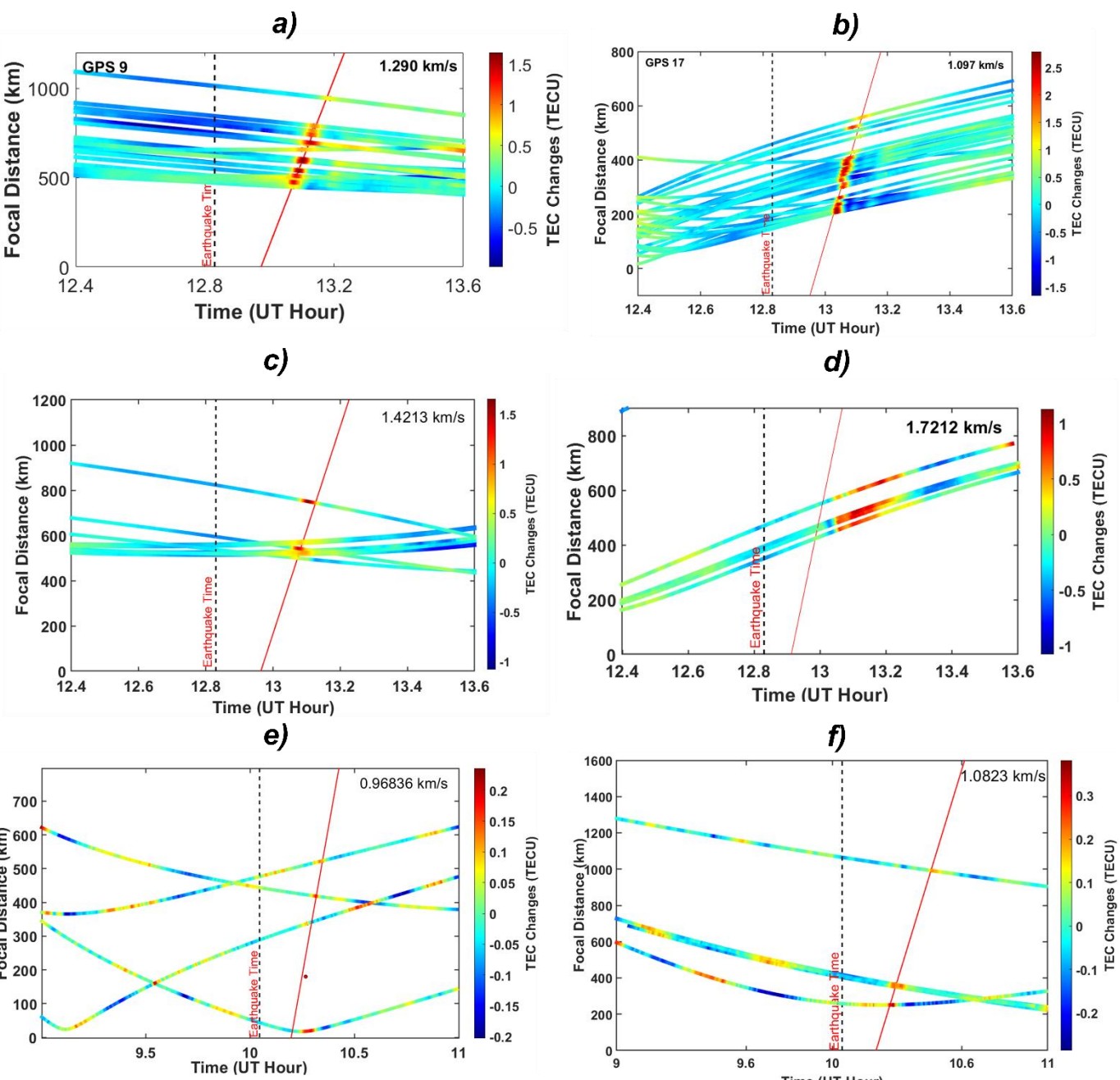

**Figure 4.** CID speed as observed from GPS PRN 9 (**a**), 17 (**b**), 23 (**c**), and GLONASS PRN 4 (**d**) of the 2016 West Sumatra earthquake is about $1.29 \pm 0.013$ km s$^{-1}$, $1.10 \pm 0.014$ km s$^{-1}$, $1.42 \pm 0.168$ km s$^{-1}$, and $1.72 \pm 0.53$ km s$^{-1}$, respectively, and also the travel-time diagram of the 2018 Palu earthquake CID based on GPS PRN 21 (**e**) and 25 (**f**) are $0.97 \pm 0.191$ km s$^{-1}$ and $1.08 \pm 0.039$ km s$^{-1}$, respectively. Distances are measured from the epicenter. The black vertical dash line indicates the occurrence of the 2016 West Sumatra earthquake (12:49 UT) and 2018 Palu earthquake (10:02:45 UT). The Pearson's correlation of CID slip as observed from GPS PRN 9, 17, and 23 and GLONASS PRN 4 are about 0.97, 0.94, 0.77, and 0.73, respectively. On the other hand, GPS PRN 21 and 25 are 0.93 and 0.99, respectively.

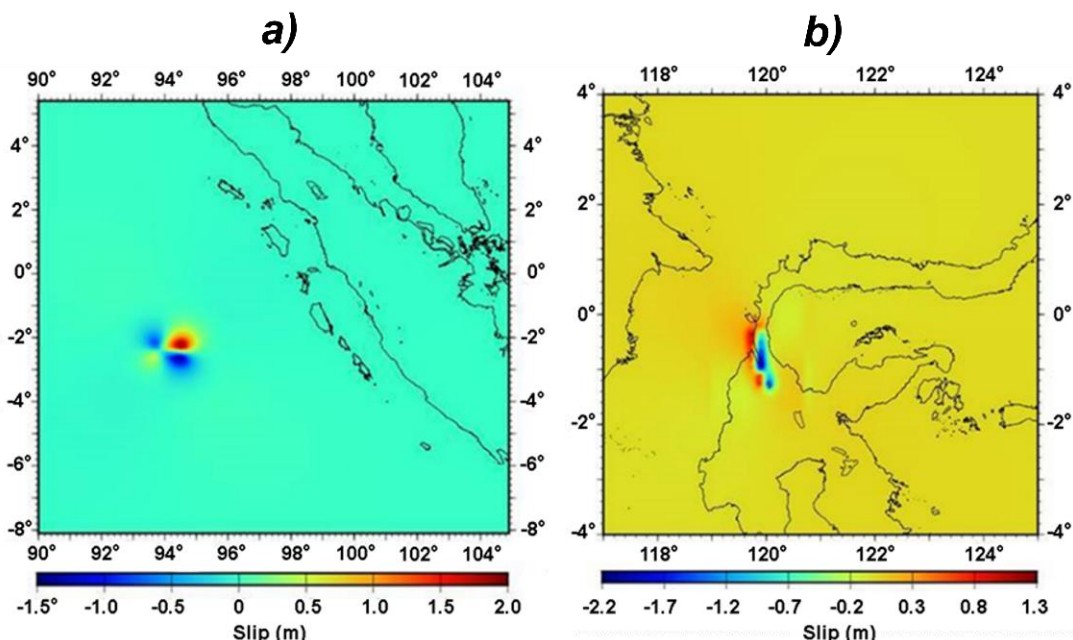

**Figure 5.** Vertical crustal movements of (**a**) the 2016 West Sumatra earthquake and (**b**) the 2018 Palu earthquake. The earthquakes were calculated using [40], using the fault geometry taken from USGS. Although the earthquakes were dominated by the strike-slip mechanism, significant vertical movements occurred.

There are two classifications of areas: buried (net gain), and landslides (net loss). Areas around Palu Bay coast tended to experience landslides. Avalanches were caused by the strike-slip earthquake at the epicenter, resulting in landslides around the bay area. The total area of landslides was 38.9 km$^2$ with a volume about 0.6 km$^3$. On the other hand, the heap area was 96.873 km$^2$ with a volume about 1.4 km$^3$. Therefore, it can be concluded that the dominant change was a heap. These seafloor deformations suggest a relationship to the activity of the Palu–Koro Fault.

## 4. Discussion

The CID amplitude of the 2016 West Sumatra earthquake was greater than the 2018 Palu earthquake. The first factor is the moment magnitude value of the 2016 West Sumatra earthquake (Mw 7.8), which was greater than that of the 2018 Palu earthquake (Mw 7.5). In addition, we used all of the satellites with no threshold or null elevation angles. This can be seen from the SIP position, which was far from the zenith station and included those with very low elevation, although low-elevation TEC data are often noisy. We used polynomial degree up to 6 to remove the effects of satellite motion at low elevations, of which CIDs with clear signals were chosen [5,16,17,19,21,22,25,37,41–45]. In this study, GPS PRN 9 and 17 (the 2016 West Sumatra earthquake) had lower angles between line-of-sight and CID wave fronts, thus producing anomalies of positive values. While the GPS PRN 25 (the 2018 Palu earthquake) had a wider angle (wide/deep angle), the positive and negative electron density anomalies tended to cancel each other [21].

The second factor is the CID directivity that propagated north and connected to the geomagnetic field in the Southern Hemisphere. The 2018 Palu and 2016 Sumatra earthquakes occurred in the Southern Hemisphere in geographic latitude, and their epicenters were located to the south of the magnetic equator. According to the International Geomagnetic Reference Field [46], geomagnetic inclination above the epicenters of the 2018 Palu main shock was −15.908° and that of the Sumatra 2018 earthquakes was −28.735°. Thus, it should have the directivity opposite to the Northern Hemisphere, that is, northward directivity in the Southern Hemisphere [16]. The SIPs of GPS PRN 9, 17, and 23 (the 2016 West Sumatra earthquake) were located north of the epicenter (Figure 2a), while the SIP

satellites GPS PRN 21 and 25 (the 2018 Palu earthquake) were in the south (Figure 2c). This makes the CID of the 2016 West Sumatra earthquake larger than the CID observed in the 2018 Palu earthquake. The relationship between magnitude of earthquake and CID relative amplitude is illustrated in Figure 6. In order to make the relationship, the CID magnitude caused by an earthquake cannot be directly compared with others, but it needs to be normalized with the VTEC value (background TEC). In this case, to obtain the background TEC values at the place and time of the CID occurrence, we used the Global Ionosphere Model (GIM). The CID of the 2016 West Sumatra earthquake was 2.9 TECU, which was normalized by 54.894 TECU (GIM) and then converted to percentage (5.3%). The same formula was applied to the 2018 Palu earthquake, with a normalized CID of 2%. This followed procedure calculations from previous research [16,42–44,47].

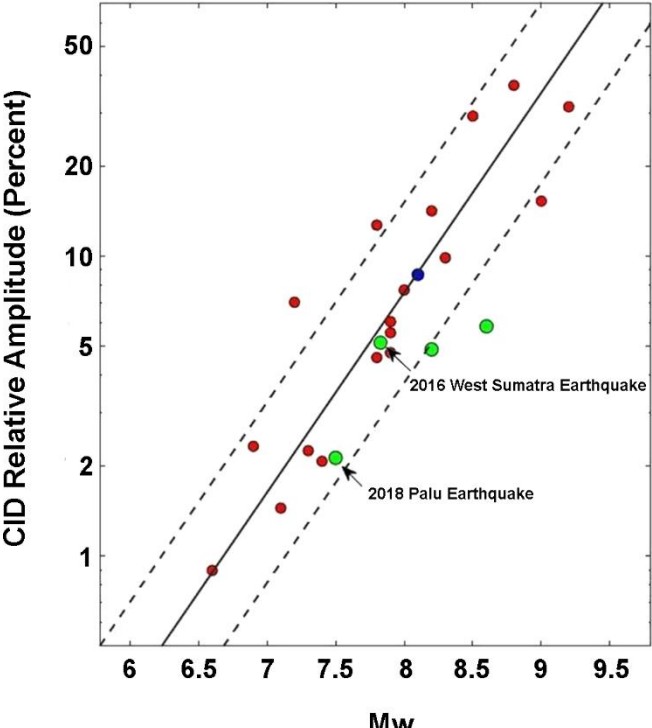

**Figure 6.** Comparison of the moment magnitude of 21 earthquakes taken from Cahyadi and Heki [16] and 2 earthquakes from this research; the amplitude of CID STEC is normalized by the background of the vertical TEC. The fault mechanism is illustrated by the symbol colors: thrust fault (red), normal fault (blue), and green (strike-slip).

The relative CID amplitude has a positive correlation, where the value will increase together with magnitude of the earthquake co-seismic crust [19]. Two dashed lines in Figure 6 assume that the observed CID observed may have a factor of uncertainty of two which marked by two dashed lines [16]. The 2016 West Sumatra and 2018 Palu earthquakes deviated negatively from the general trend, and two others that deviated negatively were the 2012 North Sumatra earthquakes (mainshock and aftershock). Smaller vertical crust movements of the strike-slip earthquake than the dip-slip event have less effect on CID amplitude, of which the deviation could have occurred.

Figure 6 shows that the 2016 Western Sumatra earthquake had a relatively larger CID amplitude compared to the 2018 Palu earthquake. This is because the 2016 Sumatra earthquake had a larger magnitude moment and uplift when compared to 2018 Palu earthquake. It is plausible to infer that the relative CID amplitude is in line with an earthquake's co-seismic crustal uplift [45]. The moment magnitude of an earthquake greatly affects the uplift component of an earthquake; if the moment magnitude is larger, the uplift will be larger as well. The strike-slip earthquake has a certain amount of vertical

crustal movement, which is about 1/5 of a dip-slip earthquake of the same magnitude [16]. It is obvious that earthquakes with larger magnitudes generate TEC perturbations of larger amplitudes [45].

## 5. Conclusions

This research examines the CID characteristics in the 2016 West Sumatra and the 2018 Palu earthquakes, including signal transmission speed (the observed velocity of 1 km s$^{-1}$, which was considered as an acoustic wave with their focal mechanism). The propagation speed (the observed velocity supported an acoustic wave origin), azimuthal asymmetry of propagation, air resonance, polarity of the first alterations, and amplitude comparison between the two earthquakes were all also investigated.

The Rayleigh and gravity signatures were absent, possibly due to the geometric alignment of the GPS network. The directivity (N–S asymmetry of propagation) of CIDs in the southern hemisphere was clearly observed, which headed to the northward direction. Resonant oscillations of the atmosphere with a frequency of ~4 mHz were found to follow the CID and last for an hour in the 2018 Palu earthquake. TEC changes similar to the 2012 Sumatra earthquake were found in the 2016 West Sumatra and 2018 Palu earthquakes, which had the same focal mechanism as strike-slip earthquakes. The TEC behavior with the same satellite-station combination over 1 day after and before for two earthquakes suggests that the observed anomaly is relevant to the earthquake.

Another important result of this study confirms the scaling law with the relationship between normalized TEC and earthquake moment magnitude. The CID on the strike-slip earthquake in the 2016 West Sumatra and the 2018 Palu earthquake has the same results on the scaling law. Based on the scaling law and tsunami arrival time in Palu Bay, possibility early warning of the tsunami can be applied for the 2018 Palu earthquake because the tsunami occurred 20–35 min after the mainshock [48], and on the other hand, the CID occurred ~13 min after the earthquake.

**Author Contributions:** Conceptualization, M.N.C. and I.N.M.; methodology, R.W.R., B.M. and S.J.; software, I.H.H.; validation, S.J. and B.M.; formal analysis, M.N.C.; investigation, I.M.A.; data curation, D.A. and D.G.P.; writing—original draft preparation, M.N.C. and R.W.R.; writing—review and editing, I.N.M. and S.J.; visualization, D.A.; supervision, M.N.C. and I.M.A.; project administration, D.G.P.; and funding acquisition, I.M.A. All authors have read and agreed to the published version of the manuscript.

**Funding:** This work was supported by Research Grant Kemitraan DRPM with project number 1310/PKS/ITS/2021, the National Natural Science Foundation of China (NSFC) Project (Grant No. 12073012), the project scheme of the Publication Writing-IPR Incentive Program (PPHKI), and the World Class Professor Program with grant number 2817/E4. 1/KK.04.05/2021.

**Institutional Review Board Statement:** Not applicable.

**Informed Consent Statement:** Not applicable.

**Data Availability Statement:** GNSS data are available from the Geospatial Information Agency (BIG) (http://inacors.big.go.id/), accessed on 2 May 2020.

**Acknowledgments:** The authors are grateful to the Geospatial Information Agency (BIG) for the GNSS and bathymetry data, the National Natural Science Foundation of China (NSFC) Project (Grant No. 12073012), the project scheme of the Publication Writing-IPR Incentive Program (PPHKI), and the World Class Professor Program with grant number 2817/E4. 1/KK.04.05/2021.

**Conflicts of Interest:** The authors declare no conflict of interest.

## Appendix A

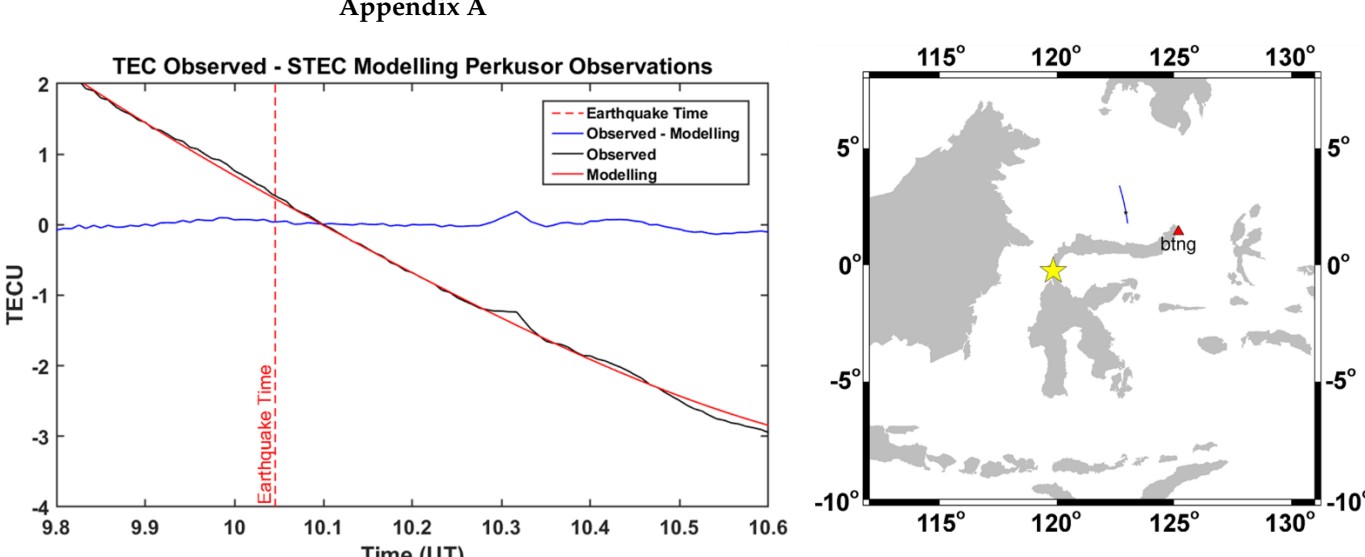

**Figure A1.** The slant of TEC changes before and after 2018 Palu earthquake measured from BTNG station using GPS PRN 21; the result is similar to that of Mikesel et al. 2019 (**left**). The variation of TEC slant (black) modelled with polynomial degree up to 6 (red). The strong positive peak shows the earthquake CID (indicated by the blue circle). The red vertical dashed lines in the time series indicate the time of the 2018 Palu Sumatra earthquake mainshock (10:02:45 UT). Furthermore, on the trajectories of GPS PRN 21 (**right**), small red stars indicate the SIP at the mainshock event. The yellow star indicates the epicenter, while the red triangle indicates GPS stations.

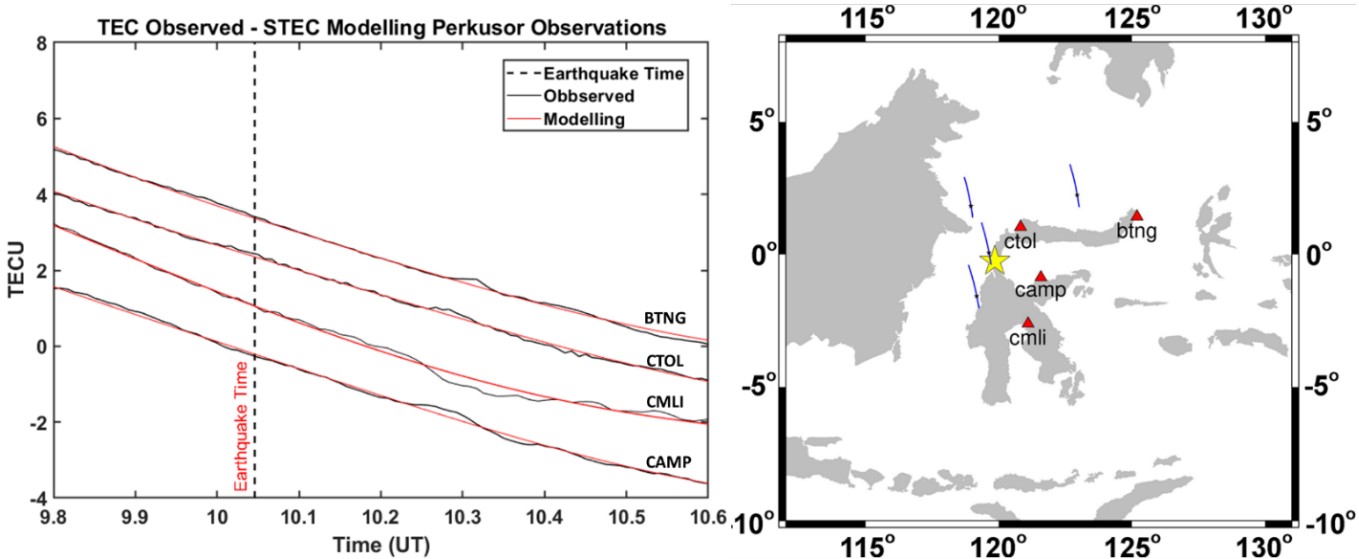

**Figure A2.** The slant of TEC changes before and after the 2018 Palu earthquake measured from CTOL, CMLI, CAMP, and BTNG stations using GPS PRN 21 (**left**). The variation of TEC slant (black) modelled with polynomial degree up to 6 (red). The strong positive peak shows the earthquake CID (indicated by the blue circle). The black vertical dashed lines in the time series indicate the time of the 2018 Palu Sumatra earthquake mainshock (10:02:45 UT). Furthermore, on the trajectories of GPS PRN 21 (**right**), small red stars indicate the SIP at the mainshock event. The yellow star indicates the epicenter, while the red triangle indicates GPS stations.

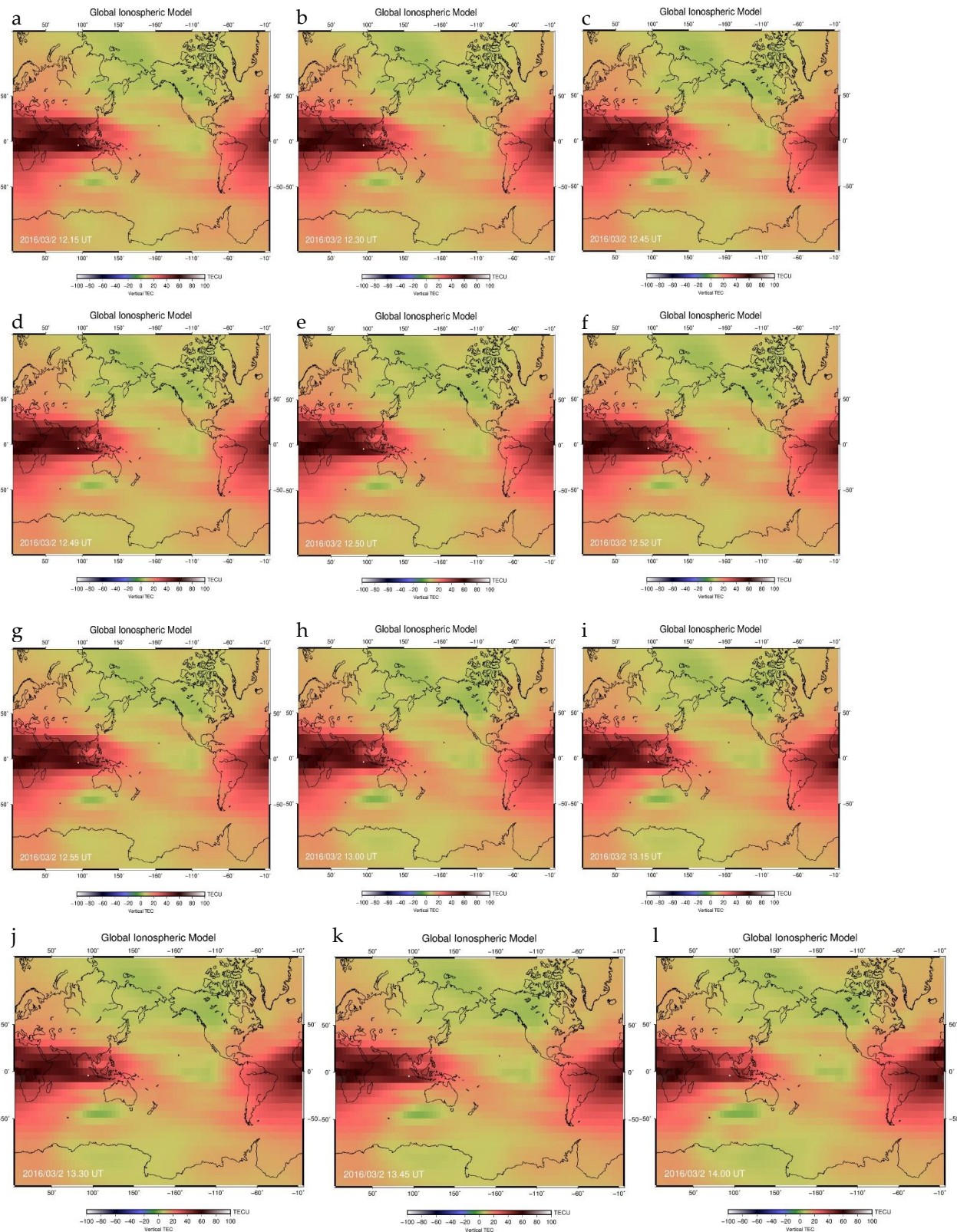

**Figure A3.** Global ionospheric maps processed during the 2016 West Sumatra earthquake on March 2, 2016 (**a–l**). The data started from ~35 min (**a**) before the earthquake to ~70 (**l**) minutes after the earthquake time (12:49:48 UT), with data intervals every 15 min. EIA propagation moves westward, which is characterized by an increase in electrons around the equatorial latitudes.

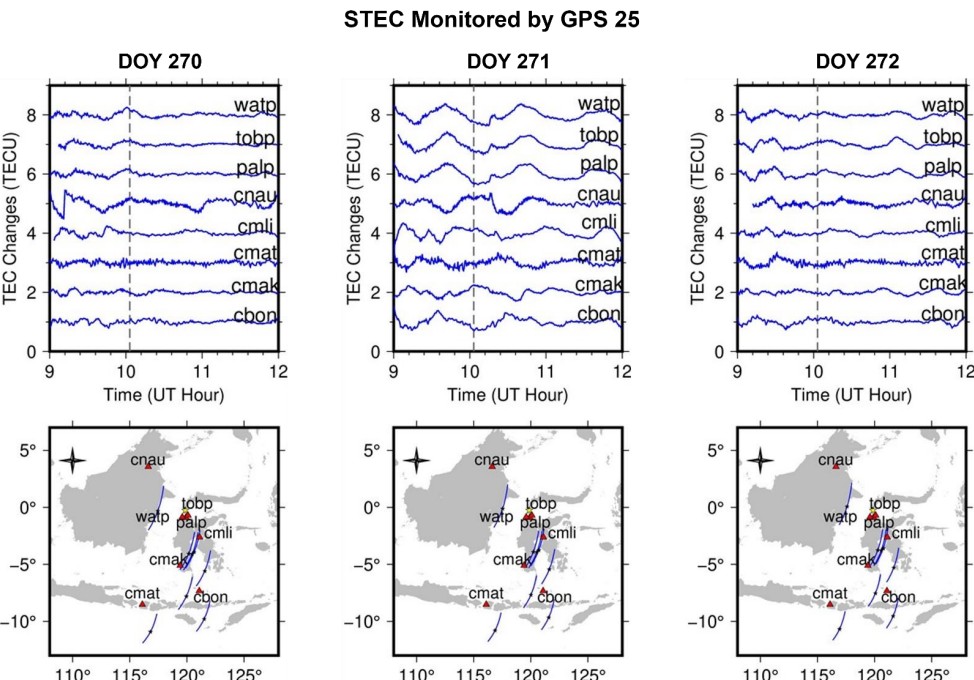

**Figure A4.** The slant changes of TEC in GPS PRN 25 one day prior and after the earthquake in which the TEC changes occurred. The black vertical dashed lines in the time series indicate the time span of the 2018 Palu Sumatra earthquake mainshock (10:02:45 UT). The small black stars indicate the SIP on the trajectories during mainshock event. The yellow stars indicate the epicenter, and the red triangle indicates the GPS stations.

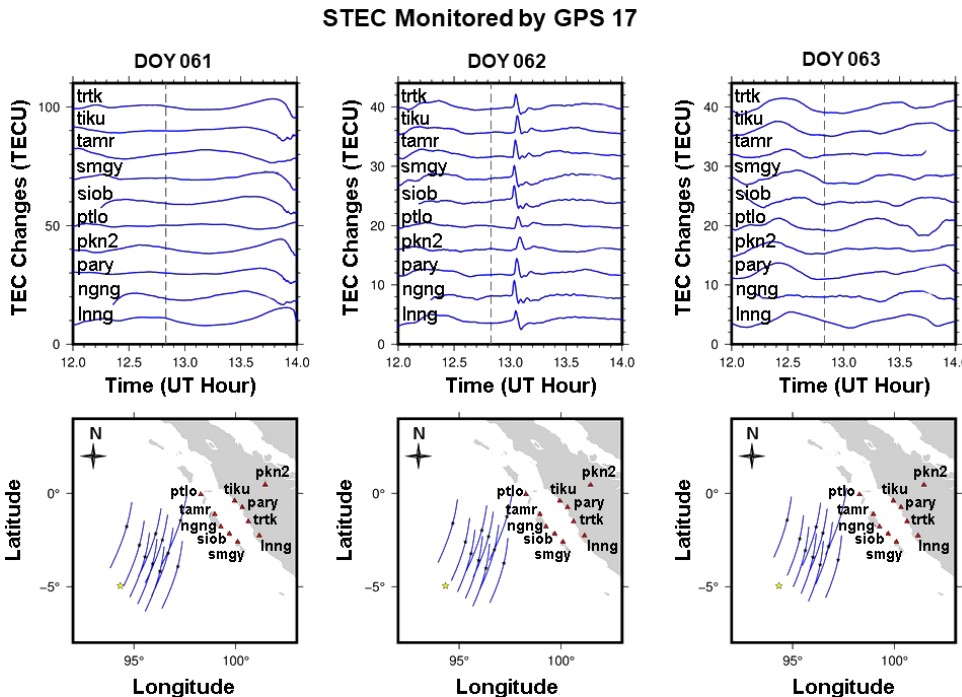

**Figure A5.** Shows the slant changes of TEC in GPS PRN 17 one day prior to and one day after the earthquake in which the TEC changes occurred. The black vertical dashed lines in the time series indicate the time span of the 2016 Sumatra earthquake mainshock (12:49:48 UT). The small black stars indicate SIP on the trajectories during the mainshock event. The yellow stars indicate the epicenter, and the red triangle indicates GPS stations.

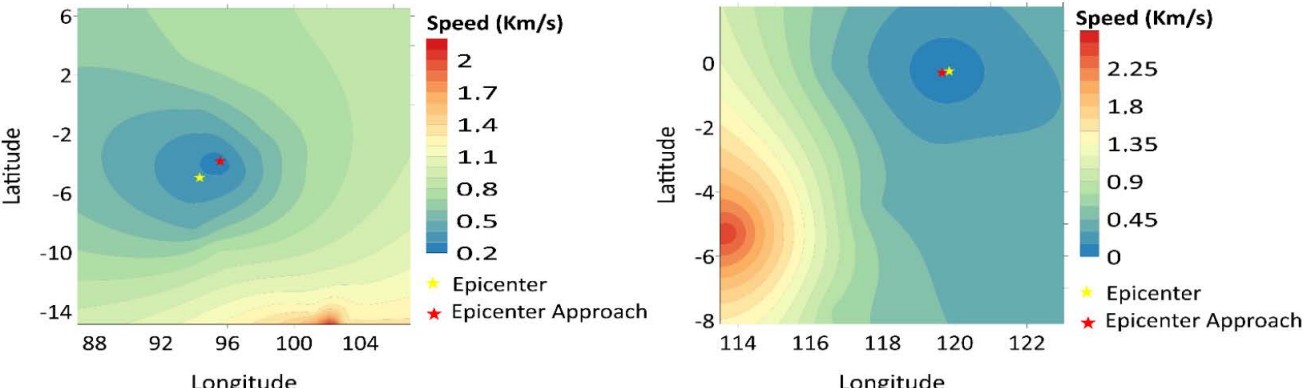

**Figure A6.** Contour of speeds for epicenter estimation with the Indonesia GPS Network: (**left**) 2016 West Sumatra, and (**right**) 2018 Palu earthquake The yellow star indicates the epicenter reported by the USGS, and the red star represent the estimated epicenter sources using Liu et al. [21] method. The earthquake epicenter measurement using INACORS shows a similar result with the modelling version.

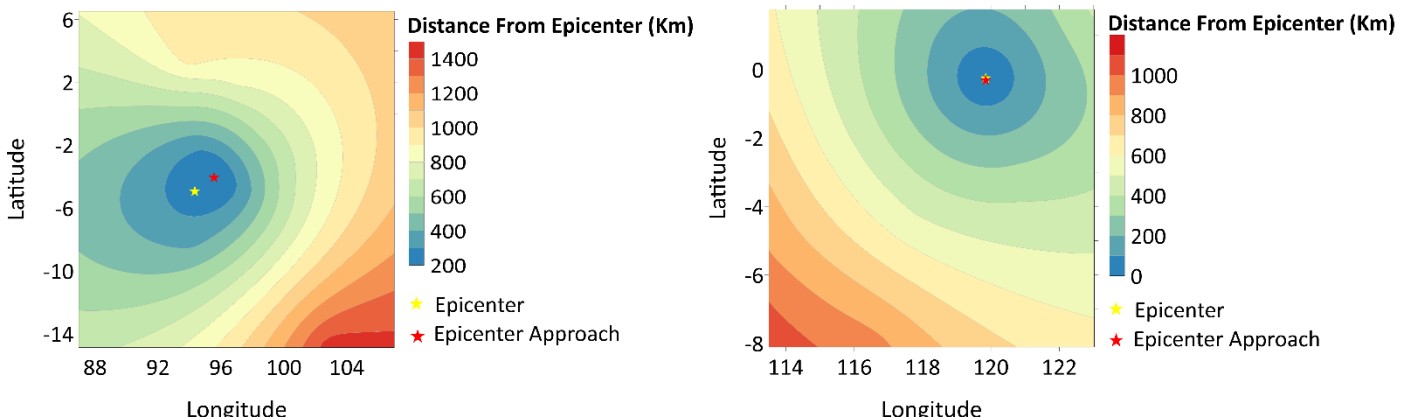

**Figure A7.** Contour of distance from epicenter estimation with the Indonesia GPS Network: (**left**) 2016 West Sumatra, and (**right**) 2018 Palu earthquake. The yellow star indicates the epicenter reported by the USGS, and the red star represents the estimated epicenter sources using Liu et al.'s [21] method. The earthquake epicenter measurement using INACORS shows a similar result with the modelling version.

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
