# Peer review of "Co-Seismic Ionospheric Disturbances Following the 2016 West Sumatra and 2018 Palu Earthquakes from GPS and GLONASS Measurements"

_remotesensing, doi:10.3390/rs14020401_

Round 1
Reviewer 1 Report
Thanks to the authors for their responses to the reviewers’ comments.
Most of my concerns were address but there are some that not addressed.
- Figure is 2a to Figure 2i show there is actually event occurrence at one point (i.e. there is a clear signature of CIDs) but it is difficult see these clear signatures in Figure 2n to Figure 2t, because there are multiple frequencies, amplitudes and phases and also significant noise present in these panels.
- Figure 3a and 3b has similar problem. That is why the spectral analysis in Figure 4b did not show any conspicuous dominant frequency at the expected time. So, for these Figures, it is generally difficult to identify an obvious event’s signature in the plasma density. Authors need to address this problems before publication.
- Lines 20 – 21, the use of “However” is weird. These are two negative word so “however” is not the right adverb.
- Line 29, the TEC amplitude of 2.9 TECU for an earthquake of Mw=7.8 and 0.4 TECU for an earthquake of Mw= 7.5 is not consistent/realistic. One seems too high and the other seems too low….. Even the Tohoku-Oki event of Mw=9.1 doesn’t amount to 2.9 TECU.
- Line 31 – 32: “….~0.87-1.65 31 km s-1 for the 2018 Palu mainshock earthquake, which are consistent with acoustic and gravity……” 87km s-1 is not consistent with gravity waves spectrum. And 1.65 31 km s-1 is not consistent with the sound speed at 250 – 350 km altitudes.
- “What is elevation cut-off angle? What is the subionic point (SIP) or IPP altitude (i.e. the peak height) considered in this study?”
I mean the authors should specify/state the cut-off elevation angle and the IPP altitude they used for their study. I didn’t asked them to define the terms as they did in their response to reviewer.
Author Response
We deeply appreciate constructive comments by the reviewers. We attach our replies to the individual comments in blue fonts.
Authors

Reviewer 2 Report
Found there are some improvements from the previous version of remotesensing-988857. In Figure 4, it is good to explain that two peaks at 3.7mHz and 4.3mHz are reasonable in comparison with other reports and not come from the alias problem.
The abbreviation of "CID" is not explained so insert the appropriate location, maybe Line 46.
Author Response

(The authors gave the same response as above.)

Reviewer 3 Report
The authors presented a study of Co-seismic Ionospheric Disturbances characteristics following the 2016 West Sumatra and 2018 Palu earthquake. CID of two earthquakes are observed using GNSS. This work is continuation of authors previous research.
The paper idea and topic is very important and interesting. However in the present form the manuscript features significant shortcomings, particularly regarding the experimentation, overall methodology and presentation. Authors should shorten the number of figures. They unnecessarily try to prove some things and bombard with some information. Therefore, the paper cannot be considered in the present form and substantial revision is required. I would suggest to rejected and to invite to resubmit after revisions before the possible publication.
Also, there are lot of technical shortcomings. I will list just a few of them
Affiliations are not listed properly.
Line 31: km s-1 => km s^{-1} and throughout the text
References should be numbered not author, year style.
What are the purposes of figures when nothing can be seen on them. They are unacceptable small and need to be fixed. Please enlarge them and insert larger text on axis.
Table 1 should be of a better design.
Line 366: “This study presented a comprehensive study…” => “This study presented a comprehensive investigation…”
Line 367: unclosed parenthesis.
figure S1 is not mentioned in the text.
All references should be written in the same way, according to the journal style.
….
Author Response

(The authors gave the same response as above.)

Round 2
Reviewer 1 Report
Thanks to the Authors for their responses to the reviewer’s comments. They were able to clarify some doubts from our previous discussions, but there are still some problems with the current version.
- Figure 3 is still very problematic.
- Figure 3a in particular doesn’t show any significant representative of earthquake signature. Looking at the behavior of delta-TEC before and after the earthquake, I cannot point out the earthquake signature in the plot for both stations as claimed by the authors.
- Also, the spectral analysis in Figure 3c doesn’t seem to be representing the time series in Figure 3a because that strong spark in Figure 3c is not present anywhere in Figure 3a.
- In the text at line 228: I don’t understand this comment: “The period is close to 4.4 mHz up to 4.8 mHz….” Is “mHz” a unit of period???
- Why are the authors not using any cut-off angle for the satellite elevation? I think this is the reason why persistent noises are present in all the lines plots shown in the manuscript.
Now, for Figure 3a, the signal to noise is very low and that is problematic.
- Lastly, the authors should be careful and verify the wave velocity obtained in this study with the time of wave arrival in the ionosphere noting the time taken and distance covered by the wave propagation. A ray-tracing technique can be applied if the source of the wave is doubtful.
Author Response
We deeply appreciated your constructive comments. We attach our replies to the individual comments in blue fonts.

Reviewer 3 Report
At first, I would like to thank the authors to have taken into account most of my comments, and to provide useful replies and changes. Now, I think that the paper provides more and significant pieces of information, making it much more scientifically sounding and relevant. However, in order to accept this paper for publication authors should make several improvements and clarifications.
The authors stated in line 335: “In order to make the relationship, the CID magnitude caused by an earthquake cannot be directly compared with others …” Can the authors further elaborate this?
It would be useful if the authors extend their statement in line 353: “Fig 6 shows that … 2018 Palu earthquake.”
Also, it is a pity that the authors did not address some things in Conclusions. Currently it is very short with very little of useful information. Conclusions should include precise, statements about the significance of the study, highlight new findings, and explain how the work could be extended in the future and should summarize the state-of-the-art of knowledge, and suggest ideas for future directions.
Also, minor remark. Authors should check, but as I can see some sub figures (e.g. a, b…) etc. are not listed in the text.
Author Response

(The authors gave the same response as above.)

Round 3
Reviewer 3 Report
Just minor remarks. References 3-10, 13, … are not in accordance to the journal style. Also, in figure S3 major tick and text are barely visible. It should be fix. No need to review the text again (in proof it can be corrected).
The paper can be accepted even in this form.
This manuscript is a resubmission of an earlier submission. The following is a list of the peer review reports and author responses from that submission.
Round 1
Reviewer 1 Report
Review: Coseismic Ionospheric Disturbances following the 2 2016 West Sumatra and 2018 Palu Earthquakes from 3 GPS and GLONASS Measurements. By Cahyadi et al.
The authors present two case studies of Earthquake/tsunami that occurred in West Sumatra on 02/03/2016 with Mw 7.8 and in Palu on 28/09/3018 with Mw of 7.5 using GNSS data.
They claim positive and negative changes in TEC at West Sumatra resulted from the position of satellite line of sight and Palu earthquake with negative TEC changes was due to the differences in coseismic vertical displacement. They found varying amplitudes, velocity and periodicity for the two-case study as a result of difference in Earthquake magnitude.
There are lots of inconsistences and miss-leading information in the manuscript. Methods are not adequately described. The language and coherence need lot of improvement. My main concerns are given below:
The abstract does not include the ‘focal mechanism’ discussed 3.3. It is not clear what the authors mean by “focal mechanisms”.
There is lack of coherent in the introduction and no sufficient background of the topic. The word “but” is used repeatedly too often. Further detail in a coherent manner is needful to understand the current state of knowledge in the field and to clarify the scientific question(s).
Abstract states Palu’s earthquake was 7.8 but introduction at line 59 states the Palu’s earthquake was 7.5.
The Abstract states that 30sec data resolution was used, but ‘data and methods’ section, line 113 states 2 min sampling frequency.
Line 96, what do authors mean by “microwave signal”? Do you mean radio wave or electromagnetic wave?
Line 97 is misleading, nowadays, GPS satellites transmit more than 2 signals.
The ‘data and methods’ section should explain in detail how the CID values were obtained.
What is elevation cut-off angle? What is the subionic point (SIP) or IPP altitude (i.e. the peak height) considered in this study?
Using higher order polyfit is problematic, Authors should show the goodness of fit and how they handle the edge effect arising from this method.
What is GPS 17 and GLONASS 4 and 9? Do the Authors mean the PRN of the satellites? Please use PRN or the SVN.
Figure 2 is show solitary wave, I expect 2 to 3 harmonic and a gradual damping of the wave as seen in other Earthquake/Tsunami induce wave studies (e.g. Manta et al., (2020), Meng et al., (2015) and Savastano et al. (2016)).
Author should provide the similar Figures (1a and b) for control days (e.g. a day before and day after the events). It will be fine to also show the TEC value itself before the polyfit is applied in data and method section.
Only one satellite is shown for GPS constellation out of 31 satellites for West Samatra. I think author should show more satellites passing at that time in similar way. Similar signal should be seemed with other passing satellites during the same period.
Also, no GPS satellites are shown for similar signal observed by the GLONASS 9 in Fig 2b for Palu’s case study, why?
Station CBLR and CBTL are not represented in 1b or 3b. Where are these stations located?
Line 135-136 “GLONASS 4 was in the southern sky, and CID amplitudes are considerably small in the stations to the north of the epicenter (Fig. 2a)” I think the opposite is the case. The amplitude looks larger in the north of the epicenter.
Fig 4 a, c, d are not mentioned throughout the text.
What is the confidence level of the spectrogram?
A time domain spectral analysis (e.g. wavelet) will be more informative over a frequency domain spectrogram, given that other figures are in time domain. Please make all scale equal.
If data resolution is 2 mins, is not possible to obtain 8.x mHz (which less than 2 min) in the spectrogram Fig 4e.
It is not clear what expertly the authors are referring to in Fig 4b. Line plot in Fig 4a and Fig 3a are noisy and their oscillation pattern is totally different from Fig 2a and b.
Line 183, do Authors mean km/s?
The IPP of GLONASS 4 is closer to the epicenter than GPS 17, why is the speed of the wave slower (see Fig 5a and b slops)?
Fig 5c, there are activities (i.e. similar amplitude) before the 1st main shock, what is responsible this?
Also, Fig 5c does not show drop in the amplitude as mentioned on line 144 -145. And the slops look the same, so it is not clear how the authors got two different velocity for GLONASS 9 as mentioned in the text.
It is not clear what the authors are trying to discuss in section 3.3 and this aspect is not mentioned in the abstract. A detail explanation of the focal mechanism that explain the positive and negative TEC change is needful.
Table 1 should come under ‘data and methods section’. I saw it much after I’ve almost finished reading the manuscript, it would have helped my reading.
Please be consistent with figure referencing. i.e. stick to either Fig or Figure throughout.
Generally, all the Figures in this manuscript are of very poor quality, need serious improvement.
The author should follow the Journal reference format.
Also, there are too many references that are either in the references list but not cited in the manuscript or cited in the manuscript but not listed in the references. Here are few examples:
Astafyeva et al. (2013). Is cited in the text but not included in the references list.
Astafyeva, E., and K. Heki. (2009 and 2011) are listed in the reference but not cited in the manuscript.
Astafyeva et al. (2014) is not cited in the manuscript but list in the references.
Afraimovich, et al. (2001) is not cited in the manuscript but listed in the references
Etc.
Author Response
We deeply appreciate the constructive comments by the reviewers. We attach our replies to the individual comments in blue fonts.
Authors

Reviewer 2 Report
The article “Coseismic Ionospheric Disturbances following the 2016 West Sumatra and 2018 Palu Earthquakes from GPS and GLONASS Measurements” presents a detailed analysis of coseismic ionospheric disturbances following two recent earthquakes in Sumatra and Palu. This study involves both ionospheric GNSS data and ground deformation observations.
The complexity of these events needs a pluri-disciplinary approach, which has been pursued by the authors. This paper presents data from a network of station, which, to my knowledge, were not yet published for the analysis of these events.
I have also some doubts on the diagrams of Figure 4, where stationary spectrograms are displayed, while the time series clearly show several different wave activities.
However, the article text needs vast improvements. The organisation of the information is not always linear. Two cases studies are presented, but their descriptions are sometimes mixed together, with some repetitions or confusing wordings.
The English text needs also improvements: many sentences are not clear, in which the subject does not correspond to the verb and complements. The scientific explanation is also at times confused and lacks to be convincing.
I therefore recommend major revisions.
The main scientific concern is that only the data from one or 2 GNSS satellites are discussed for each event. The observation geometry of CID is a key parameter to understand what has been observed. Many of the ground stations used provide both GPS and GLONASS observations, thus about 10 satellites are expected to be in view of each stations, allowing to analyse not only the earthquake signal, but the whole TID activity during these events. The radiation pattern of the earthquake source, along with a deeper discussion of the magnetic field effect at the various azimuths with respect to the epicentre is needed.This would be very helpful to understand better why only the satellites shown in this work allowed to observe a clear CID.
A second major point is related to the lack of modelisation of the propagation of waves in the atmosphere and ionosphere. See for instance the work of Mikesell et al (2019) presenting a model of the CID following Palu earthquake, in which only one IGS station (not shown in this work) was used.
Results of ionospheric CID analysis, modelisation and source location have been already published, based on GNSS stations not used in this work. I suggest citing them and discuss the results of this work in the light of these additional data:
Liu, JY., Lin, CY., Chen, YI. et al. The source detection of 28 September 2018 Sulawesi tsunami by using ionospheric GNSS total electron content disturbance. Geosci. Lett. 7, 11 (2020). https://doi.org/10.1186/s40562-020-00160-w
Mikesell, T. D., L. M. Rolland, R. F. Lee, F. Zedek, P. Coïsson, and J.-X. Dessa (2019), IonoSeis: A package to model coseismic ionospheric disturbances, Atmosphere, 10(8), doi:10.3390/atmos10080443.
Detailed remarks. Note that I did not indicate all the sentences that need English improvements. I suggest to revise carefully the whole text.
lines 39-40: this sentence does not have a verb.
line 41: the cited paper of Calais et al. (1998) does not deal with mine blasts, but earthquakes and Space Shuttle. The correct paper discussing mine blast is:
Calais, E., J. B. Minster, M. Hofton, and M. Hedlin (1998), Ionospheric signature of surface mine blasts from Global Positioning System measurements, Geophysical Journal International, 132(1), 191–202.
lines 50-60. From the text I did not understand clearly if the earthquakes for which USGS announced that it would not produce a tsunami is the same that generated a tsunamis in Palu Bay. A description of the area affected the tsunami inundation would also be helpful. Palu Bay is long and narrow, increasing the risk of tsunamis. There is still an ongoing debate on the possible ionospheric contribution for the early warning systems. In this case, the time of tsunami arrival is comparable to the one of ionospheric detection, making it very challenging.
Section 2 Data and Methods describe the details of GPS signals carrier frequencies, but do not detail the ones for GLONASS, which has also been used in this study. This system uses slightly different carrier frequencies than GPS and the satellites use each a specific frequency.
line 99 the sentence can be improved, e.g. “ For accurate positioning, the methodology used to remove the ionospheric delay is to make an ionosphere-free linear combination of…”.
line 103: typo “expressed”
line 104: the cited work of Heki (2018) is not included in the bibliography section.
line 106 : TID are named “disturbances” and not “disorder”
line 111: which is the volcano ?
line 111: There is no reference to figure 1 in the manuscript. I think it fits here for panel 1a and on line 143for panel 1b.
line 118: the angle of satellite altitude can be named “elevation angle”.
line 119: what does it mean “the penetration of the viewing line angle into the thin ionosphere”?
Lines 125-127: which is the length of the time-window over which the polynomial fit was applied? Is it the whole length of the GPS/GLONASS pass over the station?
line 139-140 the sentence “the directivity of CID propagates towards the north and attributes it to the geomagnetic file in the Southern Hemisphere” is not clear and should be reformulated.
line 142: it is not necessary to repeat the definition of the acronym INACORS. It was already defined on line 110, but with a different form of the acronym and a different explanation.
Lines 160 - 163 : Figure 3a shows data between 9 and 15 UT, while the caption indicates 9-12.
The sentence “which was recorded by some satellites is unclear”: the satellites do not record data, they transmit a signal which is recorded by a ground receiver.
Figure 4 present spectrograms but the text of the article does not explain how these spectrograms have been computed. The time window shown for each of panels c, d, e could be made evident on panel a. It is surprising to see that the spectrograms are constant throughout the time-window. Panel a shows clearly that the wave activity is not stationary in the half hour following the earthquake.
line 165: I would not call the superposition of various frequency “monochromatic”. These figures show a stationary event, but I’m really doubting that this is due to the width of the time-window for spectrogram calculation.
line 200-205 The caption of figure 5 does not explain what are the red lines (distance travelled from the epicentre at the speeds indicated in the caption?). The English sentences of the caption are not clear “Travel-time diagram …. is about 0.678 km s-1” the subject of the verb cannot be “Travel-time diagram”, but “CID speed as observed from GPS satellite 17”.
line 213-214: “relative CID value has negatively deviated from the existing trend (Fig.7)” I cannot understand this sentence. I also do not understand how Figure 7 explain this.
lines 229-238. The explanation of the right panel of Figure 7 is not clear. I do not understand if this result is part of this study or reproduced from other works. In the text there is no explanation on how this result was obtained, nor a citation of the source of this information. In particular it is not clear if the landslides occurred during this earthquake.
Acknowledgment section: I think in this section the data sources should be listed again and acknowledged.
Author Response
We would like to thank for the constructive comments by the reviewers. We attach our replies to the individual comments in blue fonts.
Authors

Reviewer 3 Report
This manuscript discusses oscillation of TEC after the 2018 Palu earthquake in the second half of subsection 3.1. The dominant frequency of oscillation is reported as 3.7 mHz and 4.4 mHz, but they are very similar with Nyquist frequency of 4.17 mHz in this case because sampling interval is 2 minutes. The peaks observed at 4.4 mHz and 3.7 mHz seem aliases each other and the frequency above 4.17 mHz in Figure 4 is meaningless due to aliasing. The frequency above Nyquist frequency should not be discussed because of the sampling theory. I recommend to move "Depth" and "Uplift" in Table 1 to the position between Epicentre and Max. CID. Locate the earthquake information other than GNSS-related data at the left side of the table, and GNSS-ralated data at the right.Author Response
We deeply appreciate constructive comments by the reviewers. We attach our replies to the individual comments in blue fonts.
Authors
